

# Learning of the user behavior structure based on the time granularity analysis model

Lin Guo and Xiaoying Liu

School of Economics and Management, Changchun University of Science and Technology, Changchun, China

## ABSTRACT

The construction of a consumption pattern can realize the analysis of consumer characteristics and behaviors, identify the relationship between commodities, and provide technical support for commodity recommendation and market analysis. However the current studies on consumer behavior and consumption patterns are very limited, and most of them are based on market research data. This method of data collection has high cost, low data coverage, and lagging survey results. The algorithm proposed in this article analyzes purchasing data from e-commerce platforms and extracts short- and long-term consumption matrices of consumers. By further processing these two matrices and removing the difference in granularity in time and marginal substitution rate, these matrices are finally integrated to form one consumption pattern matrix that can describe the characteristics of consumer consumption behavior in a period of time. Extensive experiments on various domains demonstrate that our proposed method outperforms state-of-the-art baselines on synthetic and real-world datasets.

## INTRODUCTION

E-commerce platforms have emerged as vast repositories of information, seemingly endless in their supply and spanning a broad spectrum of topics. These include diverse events, intricate interpersonal relationships, evolving consumer attitudes, and numerous other facets of modern life. These platforms have fundamentally transformed the way we shop and interact, fostering an environment where vast amounts of data are generated on a continuous basis. This data holds immense potential value, offering insights into consumer preferences, market trends, and operational efficiencies. However, the existing models utilized to analyze this data often fall short in capturing the full depth and complexity of the information available.

One significant knowledge gap in current models lies in their ability to effectively process and interpret the textual data generated on e-commerce platforms. While the volume of text data is enormous, extracting meaningful insights from it requires sophisticated methodologies. Current technologies, such as text mining, enable us to make strides in this direction by identifying keywords, phrases, and themes within the text.

Corresponding author
Lin Guo, guolin@cust.edu.cn

However, these tools often struggle with nuanced language, context-specific meanings, and the dynamic nature of consumer sentiment.

Technologies such as text mining, statistical analysis, association analysis, and visualization are integral to unlocking the potential of e-commerce data. They enable a range of analytical tasks, including emotional orientation analysis, which helps understand consumer sentiment and opinions. Information extraction techniques enable the automatic identification and extraction of relevant information from unstructured data, a task that would be otherwise impossible to manage manually. User influence analysis reveals key influencers and their impact on consumer behavior, providing valuable insights into how opinions and trends spread on these platforms.

Despite their capabilities, these analytical functions often face limitations in their current implementations. For instance, emotional orientation analysis may struggle with sarcasm, irony, or cultural nuances in language, leading to misinterpretations of consumer sentiment. Information extraction techniques can sometimes miss critical details or misclassify information due to imperfections in the algorithms or the inherent ambiguity in natural language. User influence analysis may overlook emerging influencers or underestimate the impact of less obvious forms of social influence.

The algorithm proposed in this article aims to address these limitations by constructing a consumption pattern matrix that comprehensively describes the characteristics of consumer behavior. This matrix is designed to capture not just the surface-level actions of consumers but also the underlying motivations, preferences, and trends that drive their behavior. By leveraging this matrix, people can gain deeper insights into consumer behavior, consumer characteristics, commodity correlations, and other critical aspects of e-commerce. This, in turn, can empower more informed decision-making, operational optimization, and an enhanced overall experience for both consumers and businesses on e-commerce platforms.

The remainder of this article is organized as follows. "Related Works" introduces the related work of this article. "Basic Model of Consumption Pattern" proposes the concept of the basic model of the consumption pattern. "Time Granularity Parameter Analysis" introduces the computational process of time granularity. "Consumption Trend Analysis" presents the analysis of the consumption trend. "Experiment" presents the experimental results. "Conclusions" states the conclusions of the research achievements.

## RELATED WORKS

Despite the widespread utilization of data analysis techniques by numerous companies for user targeting and product promotion, there remains a scarcity of effective methodologies for consumer behavior analysis. The study of purchase behavior has garnered significant attention from researchers spanning diverse fields like epidemiology, computer science, and sociology, who have developed various information diffusion models to capture and model the trajectories of consumer behavior.

Network swarm intelligence has witnessed remarkable progress in recent years, particularly in the realm of dynamic network analysis. This analysis encompasses several key aspects: (a) exploring the evolution of network structures and identifying emerging

trends (*Nian, Luo & Yu, 2022*), (b) investigating online learning mechanisms in complex networks (*He, Du & Zhuang, 2020*), (c) comparing the outcomes of diverse processing models and assessing their applicability across various domains, and (d) forecasting the status of individual nodes and the overall network structure (*Liu, Chang & Jia, 2022*; *Guo, Peng & Zuo, 2016*).

The social network analysis technology holds immense potential in the realm of consumer behavior data analysis, as consumer behavior data inherently possesses the characteristics of social network data. This data, when harnessed effectively, can yield invaluable insights. Here are a few key analysis methods that demonstrate this potential: (a) Strong tie detection: To understand the depth and strength of consumer relationships, strong tie detection comes into play. This method relies heavily on structural information, particularly triangular relationships, to identify closely knit groups and influential individuals within the network (*Peixoto, 2022*; *Jahani, Fraiberger & Bailey, 2022*). Such insights can aid in targeted marketing and community engagement strategies. (b) Advisor–advisee relationships: The analysis of advisor–advisee relationships, social action tracking, and other types of relationship transfers provides a nuanced understanding of how information, opinions, and behaviors propagate through consumer networks (*Mahdavisharif, Jamali & Fotohi, 2021*; *Ma, Zhang & Zeng, 2019*; *Barley, Dinh & Workman, 2022*). This is crucial for understanding the diffusion of trends, brand awareness, and consumer sentiment. (c) Topological structure analysis: The topological structure of a consumer network reveals patterns of connections and clusters that can be analyzed through techniques like normalization, modular analysis, and random data flow analysis (*Wei, Yu & Wang, 2021*). (d) Node clustering and classification: Node clustering and classification techniques allow researchers to categorize and group consumers based on their behaviors, preferences, and relationships (*Oloulade, Gao & Chen, 2022*; *Jin, Liu & Zhao, 2022*). This not only simplifies the analysis but also enables the development of more personalized and tailored marketing strategies. In summary, the social network analysis technology offers a powerful toolkit for understanding consumer behavior data, enabling businesses to make more informed decisions and develop more effective marketing strategies.

At present, there are many research methods that combine computer data analysis technology with management and economics. *Sasaki, Kawane & Miyahara (2021)* conducted a study exploring the utilization of internet search data in developing effective investment strategies. *Sifat & Thaker (2020)* and *Sun, Lv & Xue (2015)* discovered a substantial positive correlation between online search behaviors and the consumer confidence index. *Hagerty & Land (2012)* capitalized on internet search data to create a consumption intention index, outperforming traditional indices in accuracy. *Lachowska (2013)* employed web search data as a quantitative barometer of consumer confidence, revealing its remarkable predictive power for individual consumption expenditures. *Weiwei, Kangya & Guangjie (2019)* introduced HINE, a novel model for embedding user networks that preserves heterogeneous information, enabling a deeper understanding of user behavior within these networks. *Dong, Hu & Zhao (2024)* established the Extended Peer Opinion (EPO) model in a more realistic scenario. In this case, the personal behavior

of participants is not only influenced by their own thoughts, but also by the behavior of their neighbors.

Despite its prevalent application, the conventional survey sampling approach for analyzing consumer behavior exhibits notable drawbacks. These include: (1) the limited sample size, which frequently leads to homogeneity errors that constrain the broader applicability of the findings. (2) The challenge in maintaining objectivity, as respondents are prone to answering questions based on social biases rather than their genuine experiences, thus skewing the survey results. (3) The protracted timeline from data collection to publication, which results in a delay in disseminating survey outcomes, diminishing their timeliness and practical value.

To overcome the limitations posed by traditional survey sampling methods in consumer behavior analysis, we have embraced internet data analysis technology as a solution. The internet is a vast repository of information, and its users are not only consumers of that information but also active creators. This bidirectional flow of data generates a colossal amount of information on the network, which serves as a rich resource for analysis. In this article, we build a robust consumption pattern model that captures the nuances and intricacies of consumer behavior. The results of this analysis are not only detailed but also comprehensive, providing us with insights that were previously unattainable through traditional survey methods.

## BASIC MODEL OF CONSUMPTION PATTERN

For the algorithm proposed in this article, we abandon the traditional consumption pattern analysis method. This algorithm gathers data directly from the internet, analyzing user consumption patterns to gain insights. The internet consumption data encompasses details about users' online behavior trajectories and purchasing habits, providing a window into real-world consumption trends and patterns. By harnessing this data, we can gain a more nuanced understanding of consumer behavior in today's digital era.

Figure 1 offers a comprehensive illustration of the construction process for the consumption behavior structure. Utilizing the concept of time granularity fusion, consumption data with varying shopping frequencies is standardized to a uniform granularity level, allowing for the integration of data across diverse users. Upon thorough analysis of the behavior consumption matrix, we extract the short-term and long-term consumption matrices. Subsequently, these matrices undergo calculations to mitigate the effects of time granularity and discrepancies in marginal substitution rates, culminating in their consolidation into a single consumption pattern matrix. This matrix serves as a comprehensive representation of consumer behavior characteristics spanning a defined period. Fine-tuning the architecture's parameters is achieved based on operational outcomes derived from real-world data and user feedback.

$$C = A_{n \times m} \otimes Y_{q \times m} \tag{1}$$

In this equation, Matrix $C$ serves as a comprehensive descriptor of the overall consumption pattern. It encapsulates the various facets and dynamics that contribute to

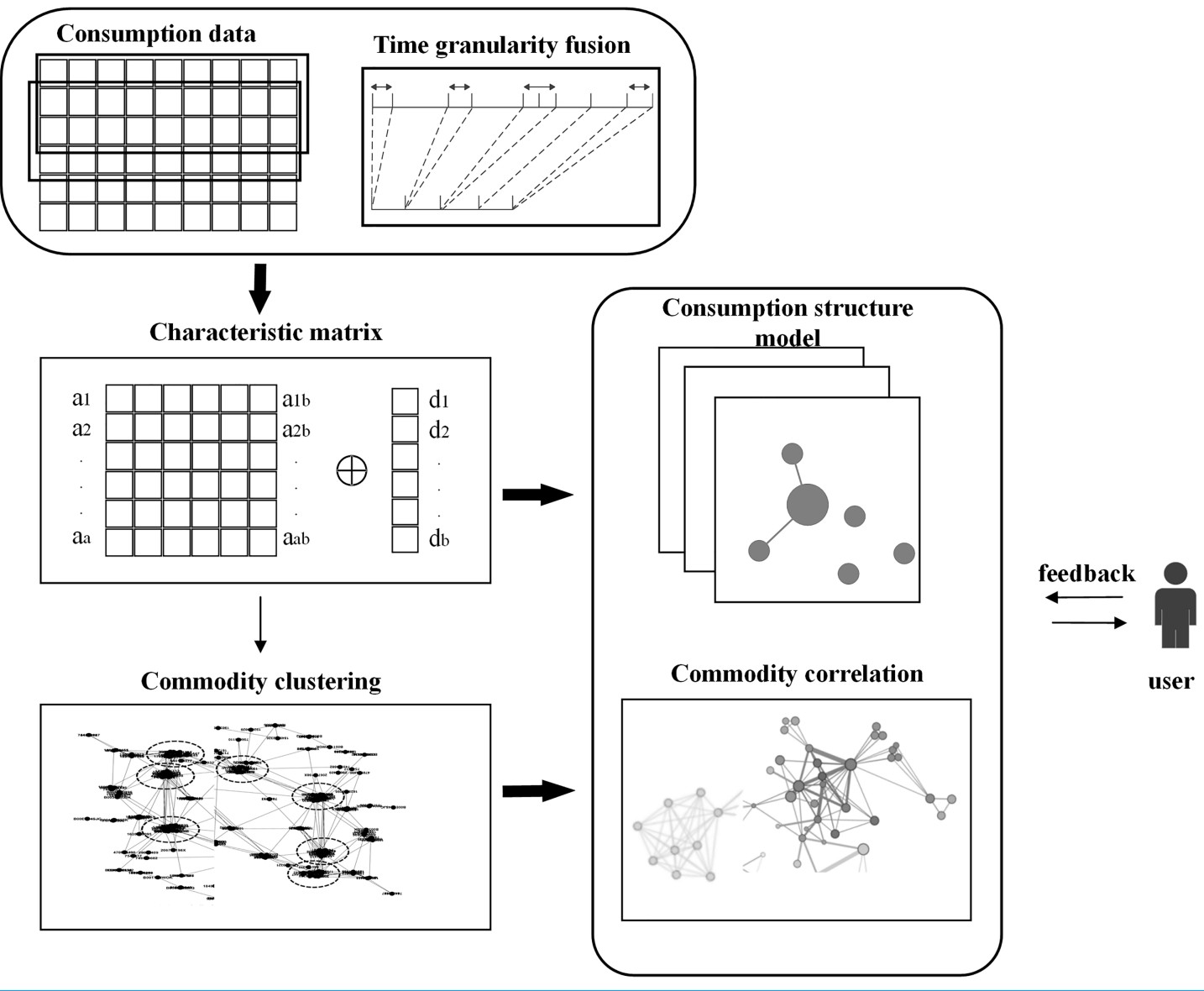

**Figure 1** **Structure of consumer behavior analysis.**

the overall spending behavior of individuals or groups. Matrix *A*, on the other hand, is dedicated to describing the data pertaining to short-term consumption behavior. This matrix captures the transient trends and patterns in spending that are often influenced by immediate factors such as seasonal promotions, short-lived fashion trends, or urgent needs. Conversely, Matrix *Y* focuses on long-term consumption behavior data. It provides insights into the enduring habits and preferences of consumers, which are less volatile and more reflective of fundamental aspects of personal or societal spending patterns. The variables *n* and *q* play crucial roles in this framework, representing the time taken to collect data for the short-term (*n*) and long-term (*q*) consumption behavior matrices, respectively.

Additionally, $m$ signifies the number of consumption categories that are being analyzed, highlighting the breadth of the study.

Equation (1) elucidates that the consumption pattern formula is composed of two integral parts: $A_{n\times m}$ and $Y_{q\times m}$. However, it is important to note that due to the inherent differences in the granularity of the data collection time for these two matrices, their respective orders (or dimensions) are inconsistent. This discrepancy poses a challenge, as it prevents the straightforward execution of an addition operation between $A_{n\times m}$ and $Y_{q\times m}$. Despite this, the symbol $\bigoplus$ in Eq. (1) is employed metaphorically to indicate that a fusion operation—rather than a simple arithmetic addition—is intended to integrate $A_{n\times m}$ and $Y_{q\times m}$ into a unified representation.

To effectively integrate Matrices $A_{n\times m}$ and $Y_{q\times m}$ into Matrix $C$, it is imperative to unify their time granularity. This process ensures that the data from both matrices are aligned in a way that allows for meaningful comparison and combination. Achieving this unification necessitates the adoption of specific execution strategies, which are meticulously introduced and elaborated upon in the subsequent chapter. These strategies are designed to address the complexities of merging data from different temporal scales, ultimately facilitating the accurate and insightful portrayal of consumption patterns within Matrix $C$.

## TIME GRANULARITY PARAMETER ANALYSIS

In the preceding section, the structural analysis algorithm overlooked the significance of time intervals, which are crucial for capturing the correlations between user actions. For instance, two behaviors occurring within a short timeframe are often interconnected, whereas behaviors separated by longer intervals may have distinct intentions.

It is imperative to note that the data collection time intervals for the short-term and long-term consumption matrices differ significantly. Consequently, to integrate these two matrices into a unified representation, it is necessary to harmonize the varying time spans. Utilizing the parameters in Eq. (1), we achieve this harmonization by fusing the time granularity $n$ (pertaining to the recent consumption behavior data collection span) and the time granularity $q$ (pertaining to the long-term consumption behavior data collection span) into a matrix with a consistent dimensional framework.

As shown in Fig. 2, Matrix $Y$ with a long time span is assigned to the same dimension as the time span of Matrix $A$. $\Delta ta_x$ represents the time interval $x$ of purchase events in $A$, $\Delta tb_x$ represents the time interval $x$ of purchase events in $Y$, and $\Delta tc_x$ represents the time interval $x$ after time span mapping, but the time intervals in $Y'$ are all equal, for example, in Fig. 2, $\Delta tc_1 = \Delta tc_2 = \Delta tc_3 = \Delta tc_4$.

The specific mapping method is shown in Eqs. (2) and (3).

$$\delta = \frac{n}{count_A} \bullet q \tag{2}$$

$$Y'_{n\times m}\{\alpha_{i\bullet}\} = \sum_j^{\delta} Y_{q\times m}\{\alpha_{j\bullet}\} \tag{3}$$

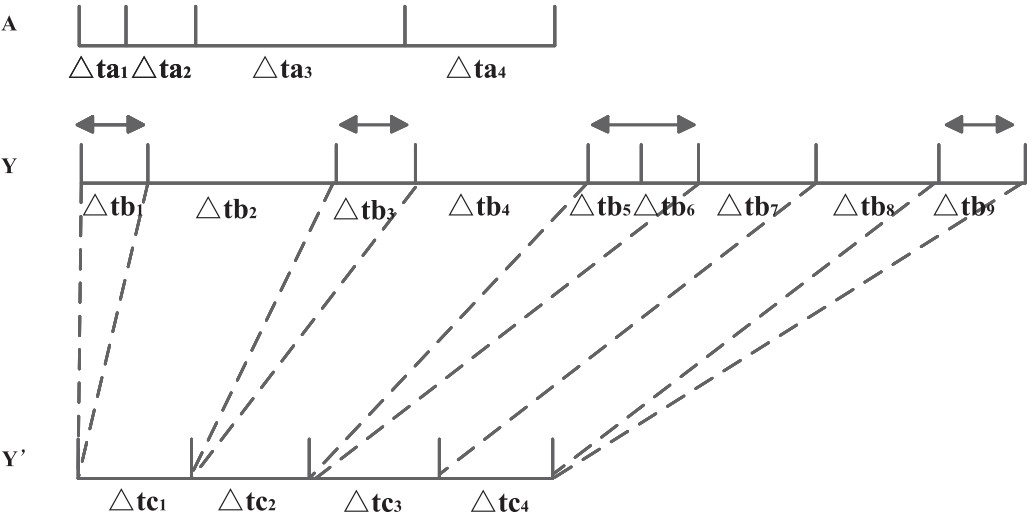

**Figure 2 Fusion process of different time granularities.**

$q$ represents the total duration covered by Matrix $Y$, encapsulating all the data pertaining to long-term consumption behavior. Similarly, $n$ signifies the total duration encompassed by Matrix A, which holds the records of short-term consumption behavior. $count_A$ specifies the precise number of purchase activities that have been documented within Matrix $A$, providing a quantitative measure of the short-term spending activity. $\delta$ denotes the granularity of time associated with Matrix $A$, referring to the specific intervals or periods over which the short-term consumption data are collected and recorded. This granularity is crucial as it determines the level of detail and resolution in the analysis of short-term spending patterns.

Equation (3) outlines the computational process for deriving a new matrix, $Y'$, from Matrix $Y$. This new matrix, $Y'$, aims to describe long-term consumption behavior by reaggregating and subdividing the consumption behavior records originally contained in $Y$, in accordance with the time granularity $\delta$ of Matrix $A$. By aligning the time granularity of $Y'$ with that of $A$, $Y'$ and $A$ become compatible in terms of their dimensions, thereby facilitating a direct addition operation between them.

With the time granularity unified, $Y'$ and $A$ can be seamlessly integrated, allowing for a comprehensive representation of both short-term and long-term consumption patterns within a single framework. Consequently, the improved consumption pattern model is presented in Eq. (4), which incorporates the refined and unified matrices to offer a more accurate and insightful portrayal of consumer behavior.

The parameter $q$ represents the total time span of the matrix $Y$, while $n$ signifies the total time span of matrix $A$. The variable $count_A$ denotes the number of purchase activities recorded in matrix $A$, and $\delta$ represents the time granularity of matrix $A$. Equation (3) outlines the calculation process for deriving the new matrix $Y'$, which characterizes long-term consumption behavior by merging and redistributing the consumption behavior records in $Y$ based on the $\delta$ time granularity. When the time granularities are unified, $Y'$

and $A$ become matrices of the same dimension, enabling the direct addition of $Y'$ to $A$. The refined consumption pattern model is presented in Eq. (4).

$$C = A_{n \times m} + Y'_{n \times m} \tag{4}$$

Equation (4), while accounting for the disparate time granularities between short- and long-term matrices, is indeed a starting point that necessitates further refinement. Consumer behavior is a multifaceted phenomenon, encompassing complexities such as marginal propensity to consume, consumption habit forgetting coefficients, and various other influencing factors. To capture these nuances accurately, Eq. (4) should be enhanced to incorporate these additional variables and dynamics. By doing so, we develop a more comprehensive and nuanced model that better reflects the intricacies of consumer behavior.

## CONSUMPTION TREND ANALYSIS

In economics, the marginal rate of substitution (MRS) represents the quantity of one good a consumer is willing to trade for another, provided that the new good offers an equivalent level of satisfaction. This economic concept serves as a valuable tool for analyzing consumer behavior for diverse purposes, as it quantifies the extent to which one good can be substituted for another. $MRS_{XY} = -\Delta Q_Y / \Delta Q_X$. MRS economics revolves around the indifference curve, a sloping graphical representation where each point signifies the quantities of good $X$ and good $Y$ that consumers are willing to exchange for each other. However, the purchase behavior of users is intricate, and accurately pinpointing the substitution relationship between two commodities using Internet data alone poses a significant challenge.

Consumption trends evolve dynamically over time. Consumers may gradually diminish their demand for specific goods, leading to alterations in their consumption patterns. Similarly, influenced by a range of complex factors, including personal preferences, social environments, and prevailing fashion trends, consumers may enhance their consumption habits. To uncover patterns in individual consumption behavior, a thorough analysis of consumers' historical consumption data is imperative.

Unlike MRS, our focus has shifted significantly from merely considering the ratio of consumption quantities between two goods to comprehensively analyzing the trends of consumption quantity changes for each individual good. In order to integrate both short-term and long-term consumption matrices effectively, we leverage consumers' consumption trends over an extended period of time. However, it's crucial to acknowledge that the process of collecting long-term data is inherently more time-consuming than gathering short-term data, which naturally leads to a greater accumulation of long-term consumption data in comparison to short-term data. If we were to add variables A and Y directly according to Eq. (4), the resultant value of $C$ would become entirely dependent on $Y'_{n \times m}$, effectively eliminating the regulatory capacity of $A_{n \times m}$ over $Y'_{n \times m}$. Therefore, it is imperative that we add $Y'_{n \times m}$ to $A_{n \times m}$ in a proportional manner to ensure the integrity and accuracy of our analysis.

The formula to calculate the consumption trend is shown below:

$$\vec{\beta} = \left\{ \beta_i | \beta_i = \frac{\sum_{p=0}^{n} Y_{n\times m}'^{N}\{y_{i,p}\} - \sum_{p=0}^{n} Y_{n\times m}'^{N-1}\{y_{i,p}\}}{\frac{1}{2}\left(\sum_{p=0}^{n} Y_{n\times m}'^{N}\{y_{i,p}\} + \sum_{p=0}^{n} Y_{n\times m}'^{N-1}\{y_{i,p}\}\right)} \times \sum_{p=0}^{n} A_{n\times m}\{a_{i,p}\}, i \in [0, m-1] \right\}$$

$$= \left\{ \beta_i | \beta_i = 2 \times \frac{\sum_{p=0}^{n} Y_{n\times m}'^{N}\{y_{i,p}\} - \sum_{p=0}^{n} Y_{n\times m}'^{N-1}\{y_{i,p}\}}{\sum_{p=0}^{n} Y_{n\times m}'^{N}\{y_{i,p}\} + \sum_{p=0}^{n} Y_{n\times m}'^{N-1}\{y_{i,p}\}} \times \sum_{p=0}^{n} A_{n\times m}\{a_{i,p}\}, i \in [0, m-1] \right\} \quad (5)$$

The vector $\vec{\beta}$ contains $m$ elements, each of which describes the consumer's consumption trend for a certain type of good. The consumption trends of various consumers are determined by comparing the difference in consumption between time $N-1$ and time $N$, which can either be positive or negative. A positive difference signifies a strong propensity for the consumer to spend on that particular category of goods, whereas a negative difference reflects a weaker propensity to consume in that category.

To add $Y_{n\times m}'$ to $A_{n\times m}$, $Y_{n\times m}'$ needs to be scaled up or down in proportion to the consumption difference at time $N-1$ and time $N$. $\sum_{p=0}^{n} Y_{n\times m}'^{N}\{y_{i,p}\} - \sum_{p=0}^{n} Y_{n\times m}'^{N-1}\{y_{i,p}\}$ is used to calculate the difference in consumption quantities. It is divided by the average of the data at moment $N$ and the data at moment $N-1$ to shrink the calculated result in proportion, and then multiplied by the total consumption amount of this kind of good in matrix $A$ to expand the consumption trend parameter according to the magnitude of matrix $A$.

According to Eq. (5), we can calculate a vector $\vec{\beta}$ which comprises m elements, based on comprehensive data derived from online consumption behavior. Each of these elements serves as a detailed descriptor of the consumer's purchasing tendency towards a specific product within the analyzed market. Subsequent to computing $Y_{n\times m}' \oplus \beta^{T}$ the resulting $Y'$ undergoes a proportional scaling process, either increasing or decreasing in magnitude. This scaled version of $Y'$ can then be seamlessly integrated into $A$, allowing for a refined and accurate representation of consumer behavior. The formula that facilitates the calculation of the consumption pattern at this juncture is articulated as follows, encapsulating the intricate interplay between $Y'$, $\vec{\beta}$ and $A$ in determining the overall consumption trends.

$$C = A_{n\times m} + Y_{n\times m}' \otimes \beta^{T} \quad (6)$$

$\otimes$ is a special matrix operation that is specified as follows.

$$H \otimes p^{T} = \begin{pmatrix} \alpha_{11} & \alpha_{12} & \cdots & \alpha_{1b} \\ \alpha_{21} & & & \\ \alpha_{31} & & & \\ & \ddots & \vdots & \\ \alpha_{a1} & & \cdots & \alpha_{ab} \end{pmatrix} \otimes \begin{pmatrix} d_1 \\ d_2 \\ d_3 \\ \vdots \\ d_b \end{pmatrix} = \begin{pmatrix} \alpha_{11} \times d_1 & \alpha_{12} \times d_2 & \cdots & \alpha_{1b} \times d_b \\ \alpha_{21} \times d_1 & & & \\ \alpha_{31} \times d_1 & & & \\ & \ddots & \vdots & \\ \alpha_{a1} \times d_1 & & \cdots & \alpha_{ab} \times d_b \end{pmatrix} \quad (7)$$

$H$ and $P$ perform the $\oplus$ operation, multiplying each row vector in $H$ by the vector $P$. This formula scales each row vector in $H$.

Once the preceding operations have been successfully executed with precision and accuracy, the short-term consumption Matrix $A$ and the long-term consumption Matrix $Y$ undergo a meticulous integration process. This integration is designed to comprehensively consider not just the variations in consumption trends across different periods but also the disparities in time frames between the two matrices, $A$ and $Y$. Furthermore, it takes into account the differences in consumption quantity baselines, acknowledging the varying scales and magnitudes of consumption activities over the short and long term. By integrating these diverse factors, the process results in the generation of matrix $C$, which serves as a comprehensive repository of both short-term and long-term consumption traits. Matrix $C$ effectively outlines the intricate and multifaceted consumption pattern of a consumer, providing a holistic view of their purchasing behavior and preferences over both immediate and extended periods.

## EXPERIMENT

The datasets used in the experiment are the following.

(1) The Amazon product dataset (*He & McAuley, 2016*; *McAuley, Targett & Shi, 2015*) comprises a vast collection of product reviews and metadata, totaling 142.8 million entries spanning from May 1996 to July 2014. This dataset encompasses various elements, including customer reviews with ratings, textual feedback, and votes on review helpfulness. Additionally, it incorporates detailed product metadata such as descriptions, categorical information, pricing, brand specifications, and image features. Moreover, the dataset also provides links that highlight related products, such as items that customers have viewed or purchased together.

(2) The EP dataset, derived from the e-commerce platform, http://www.dianping.com, encompasses a comprehensive dataset comprising 15,890,209 entries, last updated in August 2018. The data collection spans various fields, including unique shop_id, province, city, city_id, area, big_cate (primary classification), big_cate_id, small_cate (secondary classification), small_cate_id, and service_rating. Additionally, it captures detailed customer feedback through categories.

(3) The Coauthor dataset, hosted on https://www.aminer.cn/data, originates from ArnetMiner (http://www.aminer.cn/). This dynamic network comprises 1,768,776 publications spanning from 1986 to 2011, authored by 1,629,217 researchers. Each year serves as a unique timestamp, with a total of 27 timestamps across the entire dataset. At each timestamp, a connection is established between two authors if they have collaborated on at least one article in the preceding 3 years, including the current year. The collaboration between authors can be viewed as an occurrence of behavior, and by analyzing their previous collaborative behaviors, we can predict their subsequent behaviors. This approach allows us to gain insights into the patterns and trends of author collaborations, enabling more accurate predictions about future collaborations based on historical data.

(4) The Simulated data is a manually curated dataset that encapsulates the anticipated pricing preferences of 1,000 users for a particular item, along with the corresponding actual price. The structure of this dataset comprises four key components: timestamp, purchase frequency, psychological pricing expectation, and the actual price point. This dataset provides valuable insights into consumer behavior and pricing strategies.

Two sets of baseline approaches are chosen for the experiments.

1) Statistics-Driven algorithm: Once the statistical data is gathered, the distinct characteristics and intricate relationships between research objects or user behaviors become evident. The essence of data analysis using this algorithm involves several critical steps: (a) Leveraging tables, charts, and other intuitive visual representations to highlight the salient features of the data. By organizing and presenting the data in a clear and concise manner, researchers can identify patterns, trends, and anomalies in user behavior. (b) Extrapolating knowledge and insights from the sample data. This allows researchers to uncover underlying trends, correlations, and potential future behaviors based on the observed data. (c) Additionally, this algorithm can be fine-tuned and customized to specific research needs, allowing for a more tailored analysis of user behavior. Whether studying collaboration patterns in academic research or analyzing user engagement on a social media platform, the Statistics-Driven algorithm provides a powerful tool for gaining insights into user behavior and making data-driven decisions.

2) The EC-Structure algorithm (*Guo & Zhang, 2019*): is specifically designed to capture comprehensive consumption data from e-Commerce platforms. It constructs detailed consumption patterns by organizing and analyzing these data, and further examines the occurrence and trends of consumption upgrading phenomena. Through the integration of multidimensional data, such as product categories, purchase frequencies, and consumer demographics, this algorithm offers a thorough and comprehensive examination of the consumption upgrading process. Moreover, it delves into the societal implications of these changes on individual consumers, providing insights into how consumption patterns evolve and influence consumer behavior. By analyzing consumption patterns, the EC-Structure algorithm is able to identify trends in consumer behavior and grasp the underlying laws governing changes in consumption structure.

To graphically depict the consumption pattern in a concise manner, we selected 10 distinct consumption categories as our experimental subjects. The resulting consumption pattern is visually represented in Fig. 3.

Figure 3A provides a detailed illustration of the consumption pattern of a single user across various periods, highlighting the fluctuations and trends in their spending habits over time. Conversely, Fig. 3B presents a comparative analysis of the consumption patterns of multiple users within a common timeframe, offering a broader perspective on how different individuals' spending behaviors align or diverge. A notable observation from both figures is the relatively minor degree of variation in several consumption categories. This consistency can be attributed to the purchase of rigid consumer goods, which are

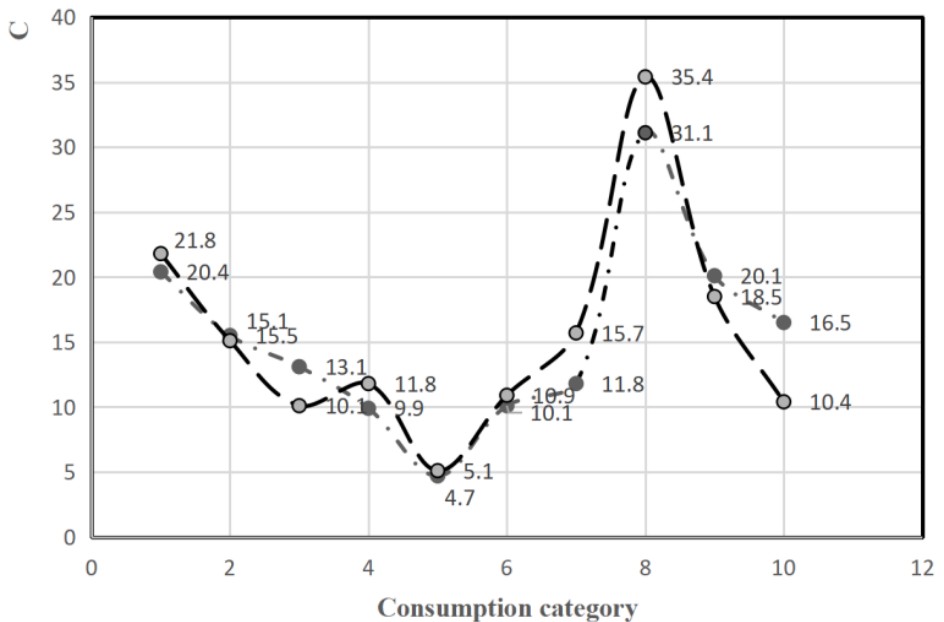

**(a) The consumption pattern of one consumer**

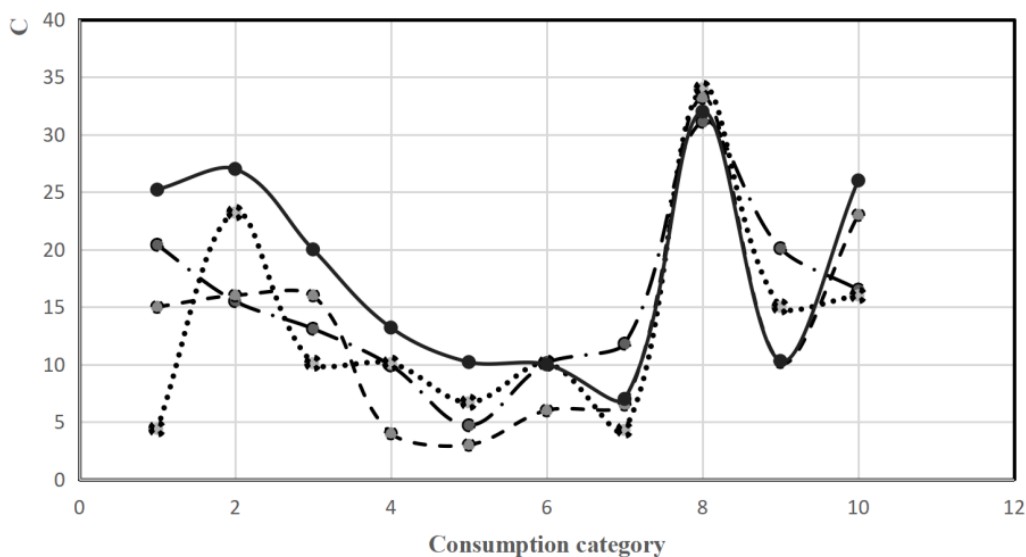

**(b)  The pattern of consumption of different consumers**

Figure 3 Consumption pattern.       

indispensable for daily living and are thus acquired in predictable quantities by each consumer. These goods, such as groceries, household essentials, and personal care products, form the backbone of consumers' spending and exhibit little variability from one individual to another. As depicted in Fig. 3, for certain identical types of goods, different consumers exhibit remarkably similar consumption trends. This convergence in spending patterns suggests that there are underlying logical relationships between consumption behaviors, transcending individual differences. These shared trends reflect fundamental

aspects of human life and societal norms that influence purchasing decisions. The figures presented in Fig. 3 serve as compelling evidence that there exist logical relationships between consumption behaviors. By analyzing these patterns, it becomes feasible to identify and discern both short-term and long-term consumption modes. By examining the data depicted in the figures, researchers and analysts can uncover insights into consumers' preferences, habits, and responses to market changes, enabling more informed decision-making and targeted marketing strategies.

The Amazon product database holds an extensive collection of data on consumer purchasing habits, yet it is interspersed with numerous irregularities. These anomalies contribute significantly to the overall computational workload without generating any new insights. Therefore, it is unnecessary to incorporate this irregular data into subsequent analytical processes. Figure 4 showcases the data pertaining to frequently observed consumer behaviors within the Amazon product dataset.

Figure 4 displays an intriguing representation of the purchase behavior data of various consumers, emphasizing the frequent and recurring structures in their purchasing patterns. This visualization provides a comprehensive snapshot of how different consumers interact with and make decisions about the products available to them. By carefully analyzing the purchase trajectories outlined in Fig. 4, one can uncover valuable insights into the connections and relationships between goods. These trajectories illustrate the sequences of purchases made by consumers, revealing patterns and trends that might not be immediately apparent from a simple list of purchased items. For instance, one might observe that consumers who purchase a particular type of smartphone often also buy specific accessories or related products.

In Fig. 5, each node is meticulously designed to signify a distinct and unique consumption category, offering a clear and concise representation of the diverse range of products and services that consumers may purchase. By carefully examining and comparing similar nodes across various consumption patterns, we can identify and achieve clustering of highly correlated goods. This clustering process involves grouping together products that are frequently purchased together or demonstrate a strong association with each other.

Tie coefficients are computed to quantify the relationships between distinct nodes. Figure 6 serves as a visual representation of consumption patterns, where the edge darkness serves as an indicator of the correlation strength between a given node and the ego node. Specifically, darker edges signify stronger correlations, and lighter edges represent weaker correlations. The numerical value displayed on each edge corresponds to the tie strength between two interconnected nodes, derived from the final computation of Matrix $C$.

To thoroughly analyze and evaluate the experimental outcomes derived from various algorithms, we utilize a comprehensive set of measurement parameters: precision (P), recall (R), and F1-score (F). $P$ is calculated using the formula $P = tp/(tp + fp)$, where $tp$ represents the number of relevant entities accurately identified, and $fp$ denotes the number of irrelevant entities incorrectly labeled as relevant. $R$ is determined by $R = tp/(tp + fn)$, with $fn$ representing the count of relevant entities erroneously overlooked.

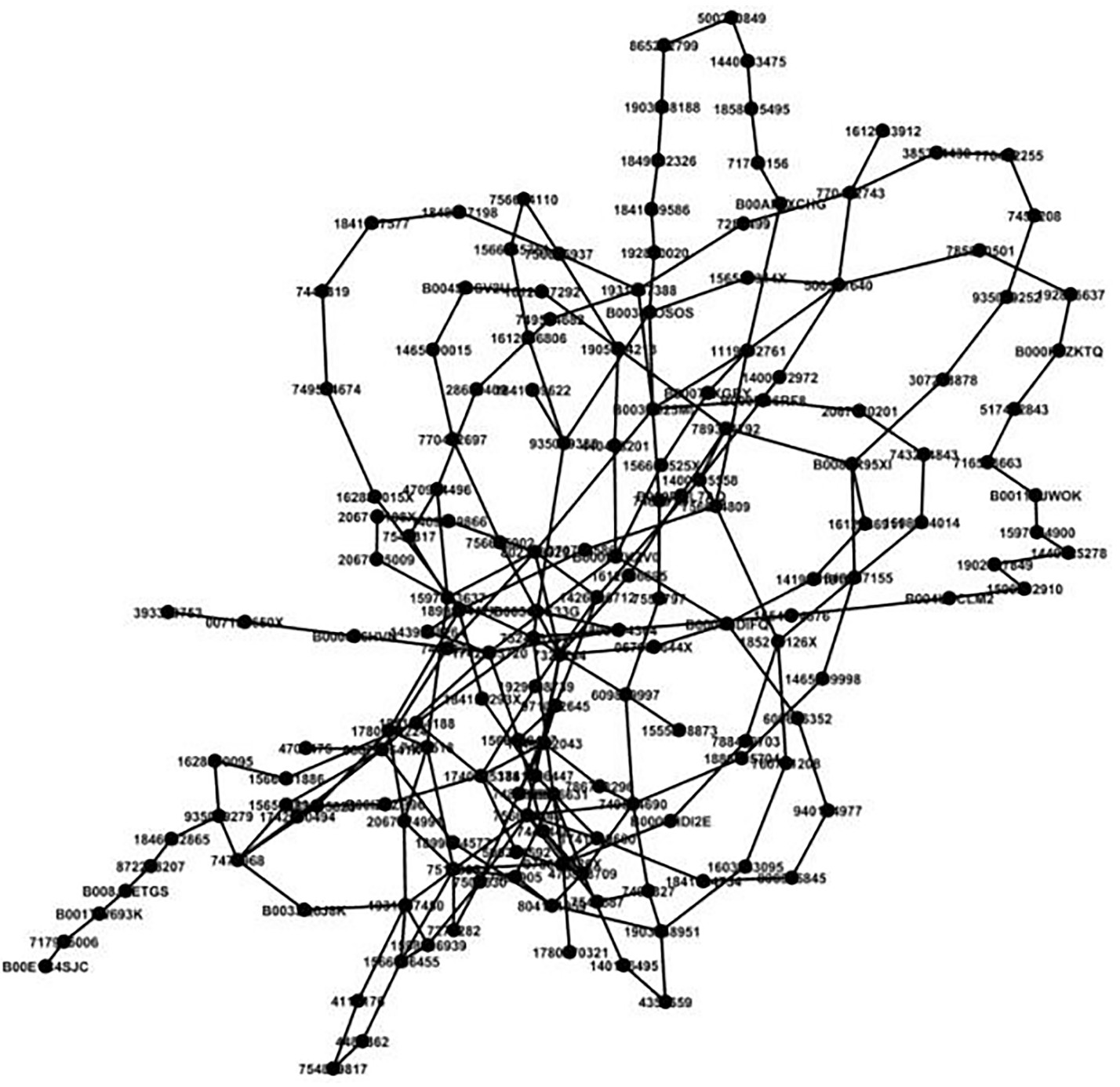

**Figure 4 Purchase behavior data that contain frequent structures.**

Finally, the $F$ is obtained through the formula $F = 2 \times P \times R/(P + R)$, offering a balanced assessment of both precision and recall.

Table 1 provides a detailed comparison of the performance of various clustering algorithms across diverse datasets. Specifically, it showcases the comparative performance of statistical algorithms, EC-Structure, and CStruc when applied to different datasets.

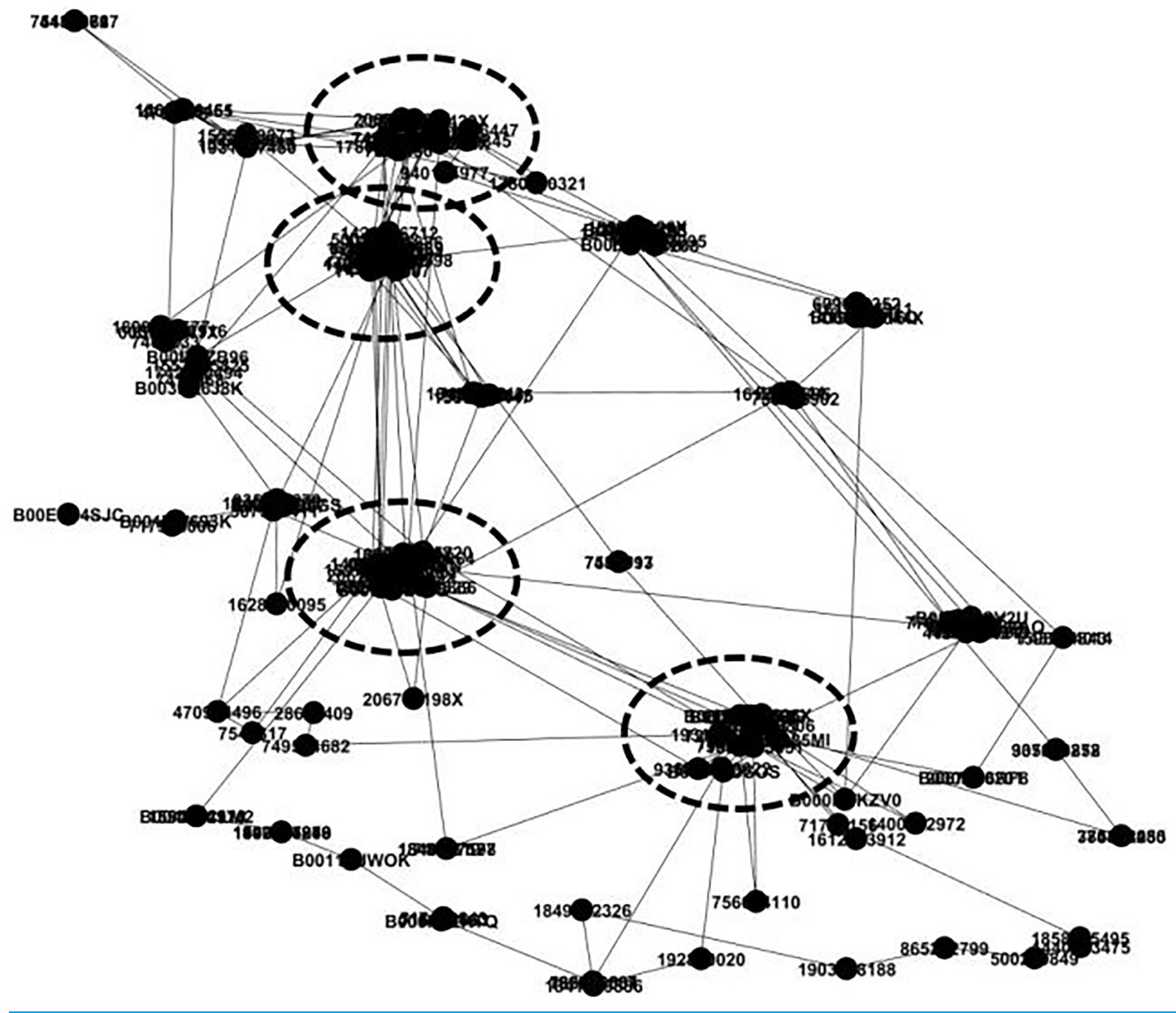

**Figure 5 Analysis of consumer goods grouping.**

The CStruc algorithm is advantageous for the EP and Coauthor datasets, as the F1-score of the algorithm is higher when using CStruc on these two datasets. On all datasets, the CStruc and EC-Structure algorithms perform relatively closely, but CStruc has a slightly higher F1-score in most cases than EC-Structure, indicating that it does a better job of balancing recall and precision. The performance of the EP dataset under the three algorithms is relatively stable and high, indicating that the dataset may have good internal structure and consistency. The Coauthor dataset achieves the highest F1-score under the CStruc algorithm, reflecting the unique structure and association patterns of the dataset

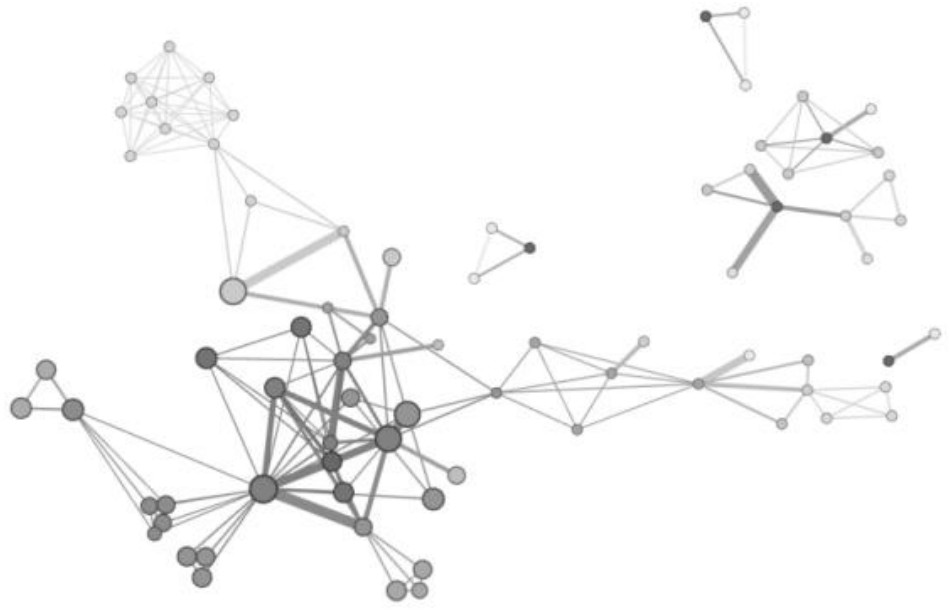

**(a) Relationships between goods**

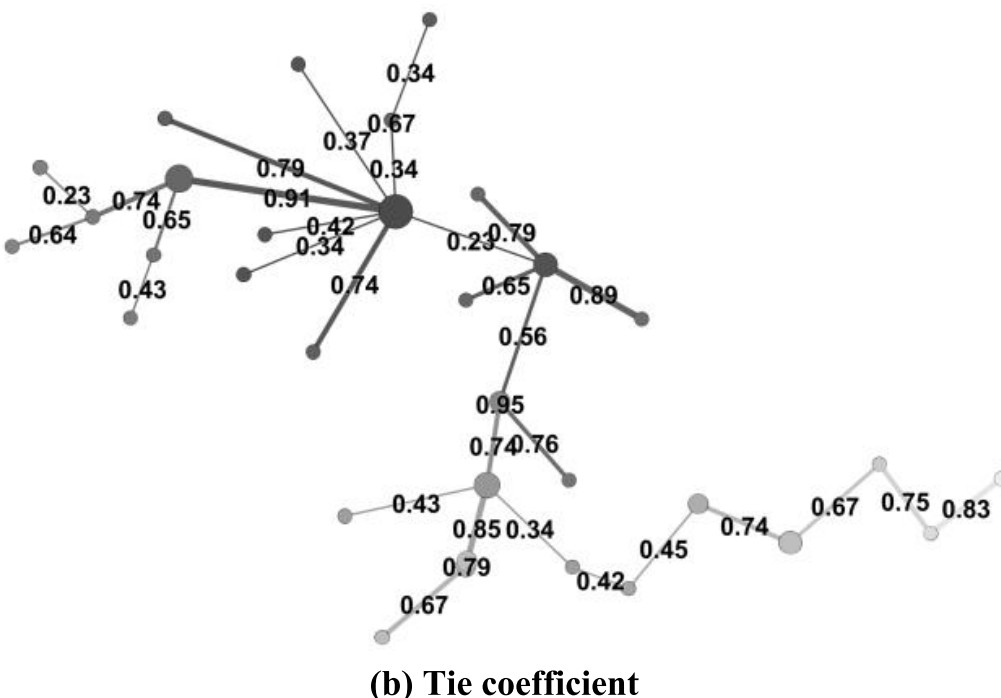

**(b) Tie coefficient**

**Figure 6** **Visual presentation of consumption pattern.**

that are particularly compatible with CStruc features. Notably, CStruc stands out as the algorithm with the most consistent execution effect and the optimal result in terms of the F1-score. This suggests that CStruc maintains stability across varying datasets while achieving high accuracy in identifying related entities and capturing all relevant instances. Therefore, CStruc emerges as a reliable and effective clustering algorithm for the given task.

Table 1 Performance comparisons of different algorithms.

| Algorithm | Statistics | | | EC-Structure | | | CStruc | | |
|---|---|---|---|---|---|---|---|---|---|
| | R | P | F | R | P | F | R | P | F |
| Amazon | 0.76 | 0.87 | 0.81 | 0.73 | 0.88 | 0.79 | 0.80 | 0.84 | 0.82 |
| EP | 0.83 | 0.88 | 0.85 | 0.86 | 0.87 | 0.86 | 0.84 | 0.90 | 0.87 |
| Coauthor | 0.80 | 0.83 | 0.81 | 0.81 | 0.84 | 0.82 | 0.85 | 0.92 | 0.88 |
| Simulated data | 0.82 | 0.83 | 0.82 | 0.84 | 0.86 | 0.85 | 0.85 | 0.88 | 0.86 |

# CONCLUSIONS

The algorithm presented in this article serves as a powerful tool for analyzing data on Internet purchase behavior, with a particular focus on uncovering both short-term and long-term consumption patterns among consumers. The key challenge lies in integrating the two distinct characteristic matrices, which often differ in terms of time granularity and marginal substitution rate. Overcoming these disparities is essential for obtaining a comprehensive consumption pattern matrix that accurately captures the characteristics of consumer behavior over a specified period.

To address this challenge, the algorithm employs a series of sophisticated techniques to harmonize the two matrices. By aligning the time scales and adjusting for marginal substitution, it creates a unified framework that facilitates the extraction of meaningful insights. The result is a consumption pattern matrix that offers a nuanced understanding of consumer behavior, revealing patterns and trends that may otherwise be overlooked.

The experiments conducted in this study demonstrate the remarkable efficacy of CStruc when compared to other similar algorithms. By utilizing multiple datasets, we were able to rigorously test the algorithm's performance in consumption pattern analysis. The results indicate that CStruc not only achieves superior accuracy but also exhibits remarkable stability across various datasets.

The algorithm presented in this article can be further refined and improved, making it a more powerful and versatile tool for analyzing data on Internet purchase behavior. The following plans are proposed for future research and development: Investigate and implement techniques to optimize the algorithm for large-scale datasets, such as parallel processing, distributed computing, and approximate algorithms. These approaches can help reduce computational complexity and runtime, making the algorithm more suitable for real-time or near-real-time analytics.

## Funding

This work was supported by the Youth Fund of Humanity and Social Science of Ministry of Education of China (Grant No. 24YJCZH079); the Project of Education Department of Jilin Province of China (Grant No. JJKH20230856SK); the Youth Program of the National

Social Science Fund of China (Grant No. 22CTQ026). The funders had no role in study design, data collection and analysis, decision to publish, or preparation of the manuscript.

### Grant Disclosures

The following grant information was disclosed by the authors:
Youth Fund of Humanity and Social Science of Ministry of Education of China: 24YJCZH079.
Project of Education Department of Jilin Province of China: JJKH20230856SK.
Youth Program of the National Social Science Fund of China: 22CTQ026.

### Competing Interests

The authors declare that they have no competing interests.

### Author Contributions

- Lin Guo conceived and designed the experiments, performed the experiments, analyzed the data, performed the computation work, prepared figures and/or tables, authored or reviewed drafts of the article, and approved the final draft.
- Xiaoying Liu performed the experiments, performed the computation work, authored or reviewed drafts of the article, and approved the final draft.

### Data Availability

The AMiner dataset is available at: https://resource.aminer.org/data.

The Amazon product co-purchasing network metadata is available at: https://snap.stanford.edu/data/amazon-meta.html.

The Sunnybrook Preclinical Simulated Cardiac EP dataset is available at GitHub: https://github.com/WrightGroupSRI/ep-dataset.

### Supplemental Information

Supplemental information for this article can be found online at http://dx.doi.org/10.7717/peerj-cs.2573#supplemental-information.

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
