# Peer review of "Learning of the user behavior structure based on the time granularity analysis model"

_PeerJ Computer Science, doi:10.7717/peerj-cs.2573_

## Round 0.1 · original submission · Major Revisions

Ddear authors,

Your article has not been recommended for publication in its current form. However, we do encourage you to address the concerns and criticisms of the reviewers and resubmit your article once you have updated it accordingly.

Before submitting the revised paper, please improve the clarity of the explanations and provide more intuitive interpretations of the results for readers unfamiliar with advanced mathematical modeling. Reviewer 2 has requested that you cite specific references. You may add them if you believe they are especially useful and relevant.

Best wishes,

Reviewer 1 ·

Basic reporting

The problem addressed—traditional consumer behavior studies' limitations due to high costs and limited data is well-established. The authors present their new algorithm and emphasize its benefits over state-of-the-art methods. The literature review is thorough, citing relevant studies on consumer behavior, data analysis, and network swarm intelligence.
However, the language could be improved for better readability. Some sections would benefit from a clearer explanation, particularly for an audience unfamiliar with the mathematical models involved.
Formulas 1, 2, and 3, could benefit from clearer explanations. Please provide more detailed, step-by-step guidance for readers who may not be familiar with these concepts, especially on how time granularity is harmonized.
explain what is being depicted in the consumption patterns shown in Figure 3 and how it relates to the experiment.

Experimental design

The experimental design is robust, and the research question is well-defined.
additional explanation regarding the parameters used in the time granularity fusion (e.g., in Formula 3) would make the method more accessible to readers outside the immediate domain.
The use of both synthetic and real-world datasets, such as the Amazon product dataset, provides strong empirical backing for the model’s performance. It is particularly encouraging that the authors address a variety of domains, showcasing the algorithm’s general applicability.
explain how the Coauthor dataset, which is non-commercial, fits into the context of consumer behavior analysis.
Provide more details about how the Statistics-Driven and EC-Structure algorithms work and how they compare with the proposed method.

Validity of the findings

The results demonstrate that the proposed model outperforms existing methods, especially in terms of the F1-score. The precision and recall values show a good balance between accurately identifying relevant consumption behaviors and avoiding false positives.
The authors link the findings back to their original research question by showing how the new algorithm effectively integrates short- and long-term behaviors to form more accurate consumer behavior matrices. However, more attention could be given to discussing potential limitations, such as scalability to larger datasets or varying consumer behavior types that may not fit the model as well.

Additional comments

The manuscript would benefit from improving the clarity of its explanations, particularly regarding the more technical mathematical aspects. T

Reviewer 2 ·

Basic reporting

Abstract
• The abstract elaborates on the problem and the solution the authors are trying to propose. However, this text would still do better with more current 2024 references to strengthen this premise of an enhanced method. Furthermore, the language could be adjusted for clarity. For example, instead of writing, "However, the current studies on consumer behavior and consumption patterns are minimal," it can be rewritten in a less absolute term since there are numerous studies.
• The authors should include recent publications and refine phrasing for clarity.
• The authors also need to rewrite the abstract to follow the following structure: background of study, problem statement, objectives, materials and methods, results and discussion, conclusion, and recommendations.
Introduction
• The introduction has set a very informative context about e-commerce platforms and consumer data. However, the transition to the proposed model is rather abrupt. Before introducing the new algorithm, the existing models' knowledge gap and specific shortcomings should be detailed.
• Expanding the literature review and better defining the research gap this paper intends to fill.
Related Works
• The related works section is informative but could have more discussion about how each referenced work relates to the authors' approach. Moreover, the references are outdated; none are from 2024 or 2023. There is a need to make the comparison with the existing models more direct.
• Update references with more recent studies from 2024 and provide clearer connections between existing literature and current research. Hence, I have suggested some recent articles relating to the study that the author should use, cite, and reference.
• Donta, P. K., Xu, A., & Li, Y. (2024). Marketing Decision Model and Consumer Behavior Prediction With Deep Learning. J. Organ. End User Comput., 36(1), 1-25. doi: 10.4018/JOEUC.336547
• Xu, Y., Chen, H., Wang, Z., Yin, J., Shen, Q., Wang, D.,... Hu, X. (2023). Multi-Factor Sequential Re-Ranking with Perception-Aware Diversification. Paper presented at the KDD '23, New York, NY, USAfrom https://doi.org/10.1145/3580305.3599869
• Yang, J., Yang, K., Xiao, Z., Jiang, H., Xu, S.,... Dustdar, S. (2023). Improving Commute Experience for Private Car Users via Blockchain-Enabled Multitask Learning. IEEE Internet of Things Journal, 10(24), 21656-21669. doi: 10.1109/JIOT.2023.3317639
• Li, T., Hui, S., Zhang, S., Wang, H., Zhang, Y., Hui, P.,... Li, Y. (2024). Mobile User Traffic Generation Via Multi-Scale Hierarchical GAN. ACM Trans. Knowl. Discov. Data, 18(8), 1-19. doi: https://doi.org/10.1145/3664655
• Wang, X., Seyler, B. C., Chen, T., Jian, W., Fu, H., Di, B.,... Pan, J. (2024). Disparity in healthcare seeking behaviors between impoverished and non-impoverished populations with implications for healthcare resource optimization. Humanities and Social Sciences Communications, 11(1), 1208. doi: https://doi.org/10.1057/s41599-024-03712-z
• Zhu, C. (2023). Research on Emotion Recognition-Based Smart Assistant System: Emotional Intelligence and Personalized Services. Journal of System and Management Sciences, 13(5), 227-242. doi: 10.33168/JSMS.2023.0515
• Amin, S., Shahnaz, M., & Mukminin, A. (2024). The impacts of the servant leadership on the innovative work behaviors: Looking at the role of public service motivation and employee engagement. Journal of Chinese Human Resources Management, 15(2), 41-54. DOI: 10.47297/wspchrmWSP2040-800502.20241502.
• Dong, J., Hu, J., Zhao, Y., & Peng, Y. (2024). Opinion formation analysis for Expressed and Private Opinions (EPOs) models: Reasoning private opinions from behaviors in group decision-making systems. Expert Systems with Applications, 236, 121292. doi: https://doi.org/10.1016/j.eswa.2023.121292
• Li, T., Li, Y., Zhang, M., Tarkoma, S., & Hui, P. (2023). You Are How You Use Apps: User Profiling Based on Spatiotemporal App Usage Behavior. ACM Trans. Intell. Syst. Technol., 14(4). doi: 10.1145/3597212
• Li, T., Hui, S., Zhang, S., Wang, H., Zhang, Y., Hui, P.,... Li, Y. (2024). Mobile User Traffic Generation via Multi-Scale Hierarchical GAN. ACM Trans. Knowl. Discov. Data. doi: https://doi.org/10.1145/3664655
• Gu, X., Chen, X., Lu, P., Lan, X., Li, X.,... Du, Y. (2024). SiMaLSTM-SNP: novel semantic relatedness learning model preserving both Siamese networks and membrane computing. The Journal of Supercomputing, 80(3), 3382-3411. doi: https://doi.org/10.1007/s11227-023-05592-7
• Ding, J., Chen, X., Lu, P., Yang, Z., Li, X.,... Du, Y. (2023). DialogueINAB: an interaction neural network based on attitudes and behaviors of interlocutors for dialogue emotion recognition. The Journal of Supercomputing, 79(18), 20481-20514. doi: 10.1007/s11227-023-05439-1
Basic Model of Consumption Pattern

• The structure in this section is fine, but the mathematical notation should be better described since not all readers may know the formulae utilized. Explaining what variables represent in the formula will make it even better. Also, a discussion needs to be done on scalability and efficiency regarding the algorithm.
• Then, provide a better context for mathematical formulations and resolve potential scalability issues.
Time Granularity Parameter Analysis
• While this section does quite well in explaining how time granularity is handled, more needs to be said to the practical aspects of the choices made for the different time intervals. Alternatively, additional case studies or examples of real-world applications may enhance the discussion further.
• Case studies or examples of real-life cases should be provided to illustrate the practical implications of the proposed approach.
Consumption Trend Analysis
• MRS is well explained; however, the section requires deeper analysis for the added value of the trend analysis compared with the current models. There is little discussion on the business applications or practical scenarios where this analysis can be applied.
• Comparing the proposed trend analysis to other methods in greater detail, discuss some practical applications.
Experiment
• Experiments are not elaborated in detail. Comparisons with other algorithms exist; however, a greater explanation regarding the choice of a dataset is needed, and the algorithm's performance across different real-world scenarios is poorly elaborated. No recent datasets from 2024 were used; rather, the used datasets appear somewhat older.
• Explain why the datasets are chosen and add experiments done with more updated data from 2024.
Results and Figures
• The figures are clear and relevant. However, the discussion around Figure 3 could be expanded to provide more insight into the significance of the findings. Also, the visual representation could include additional metrics to highlight the strengths and weaknesses of the proposed method.
• Expand the discussion on the findings and consider adding more visual metrics to clarify performance comparisons.
Conclusion
• The conclusion summarizes the findings well, but it can be even more robust regarding novelty and impact. Also, there is no discussion on the limitations of this study and which part might be worked on in the future.
• These should include highlighting the work's unique contributions, discussing limitations, and identifying future directions for research.
References
• The references are somewhat outdated, with none from 2024. This weakens the argument that the proposed method is relevant and up-to-date with recent developments in the field.
• Update the references to include more recent works, particularly from 2024, to reinforce the relevance of the research.

Experimental design

• Experiments are not elaborated in detail. Comparisons with other algorithms exist; however, a more significant explanation regarding the choice of a dataset is needed, and the algorithm's performance across different real-world scenarios is poorly elaborated. No recent datasets from 2024 were used; instead, the used datasets appear somewhat older.
• Explain why the datasets are chosen and add experiments done with more updated data from 2024.

Validity of the findings

No comment

Additional comments

No comment

---

## Round 0.2 · accepted · Accept

Dear Authors,

Thank you for addressing the reviewers' comments. Your manuscript now seems ready for publication.

Best wishes,

Reviewer 1 ·

Basic reporting

The text, ideas, concepts and solving are well written.

Experimental design

The methods are described with sufficient details.

Validity of the findings

The findings are well presented.

Additional comments

No additional comments.

Reviewer 2 ·

Basic reporting

The authors have adequately addressed most of the concerns raised in the initial review, resulting in significant improvements to the manuscript. They have responded thoroughly to each comment, clarifying and expanding on key points as requested. The revisions strengthen the research's clarity, rigour, and presentation, demonstrating the authors' commitment to enhancing the work. I am confident that the manuscript now meets the journal's high standards and recommend it for acceptance.

Experimental design

No comment

Validity of the findings

No comment

Additional comments

No comment